# Appropriate Buffers for Studying the Bioinorganic Chemistry of Silver(I) [†]

**Lucille Babel *** [ID]**, Soledad Bonnet-Gómez and Katharina M. Fromm ***

Chemistry Department, University of Fribourg, Chemin du Musée 9, 1700 Fribourg, Switzerland;
sole82@bluewin.ch

*   Correspondence: lucille.simond@unifr.ch (L.B.); katharina.fromm@unifr.ch (K.M.F.);
    Tel.: +41-2630-087-32 (K.M.F.)

† Dedicated to the radical chemist Prof. Bernd Giese on behalf of his 80th birthday.

**Abstract:** Silver(I) is being largely studied for its antimicrobial properties. In parallel to that growing interest, some researchers are investigating the effect of this ion on eukaryotes and the mechanism of silver resistance of certain bacteria. For these studies, and more generally in biology, it is necessary to work in buffer systems that are most suitable, i.e., that interact least with silver cations. Selected buffers such as 4-(2-hydroxyethyl)-1-piperazineethane sulfonic acid (HEPES) were therefore investigated for their use in the presence of silver nitrate. Potentiometric titrations allowed to determine stability constants for the formation of (Ag(Buffer)) complexes. The obtained values were adapted to extract the apparent binding constants at physiological pH. The percentage of metal ions bound to the buffer was calculated at this pH for given concentrations of buffer and silver to realize at which extent silver was interacting with the buffer. We found that in the micromolar range, HEPES buffer is sufficiently coordinating to silver to have a non-negligible effect on the thermodynamic parameters determined for an analyte. Morpholinic buffers were more suitable as they turned out to be weaker complexing agents. We thus recommend the use of MOPS for studies of physiological pH.

**Keywords:** silver; buffer; association constant; HEPES

## 1. Introduction

A well-known list of buffers was published between 1966 and 1980, called Good's buffers, for their use in biological systems [1]. This list contains essentially sterically hindered amines that aim to replace common buffers used in biology such as imidazole, sodium phosphate and sodium citrate. Indeed, these previously employed buffers are inadequate for certain experiments because of their reactivity towards small molecules (ATP), metal ions, or because of their toxicity for the cells [2–8]. For example, a phosphate buffer leads to precipitates with many cations and is known to inhibit or enhance certain reactions of a cellular system [2,3]. Imidazole is a very good complexing ligand for many metal cations and, due to its similar structure, could replace histidine residues in metal binding proteins [9–11]. Good's buffers on the contrary were believed to be largely inactive towards the cell metabolism and thus should not interact with any biological molecule and/or metal ions. Nevertheless, since this list was established, many studies have proved that most of these sterically hindered tertiary amine-based buffers are able to coordinate slightly some metal ions [12,13]. Therefore, binding constants determined for other ligands could be affected by the presence of these buffers, which are usually in large excess compared to the ligand to ensure a stable pH, hence it is a necessity to know these values. A correction can then be applied to the thermodynamic model to take into consideration the effect of the buffer. To limit the effect of this correction, careful consideration of the metal ions in solution and the concentration of the buffer is necessary prior to use. For example,

complexation of copper(II) by buffers was thoroughly studied over recent years and it was shown that Good's buffers coordinate the metal ion with variable but non-negligible affinities of $3 \leq \log K_{Cu,L} \leq 5$ [14–16]. However, most of the studies found in the literature concern divalent metal cations and little is known on monovalent ones [17,18]. Moreover, publications on the morpholinic and piperazinic family of buffers are sometimes concluding to contradictory results [12].

Our group is interested in the use of silver as an antimicrobial agent. Silver is used in in vitro studies to investigate e.g., the silver resistance mechanism of some bacteria or in studies investigating toxicity and/or antimicrobial properties of silver agents, yet appropriate buffers for this kind of studies are lacking in the literature. We have recently been studying peptide models inspired by the protein SilE, a protein of the silver efflux pump in Gram negative bacteria, which is able to bind a large amount of silver(I) [19,20]. In this case, phosphate buffer could not be used because of the immediate formation of the poorly soluble silver phosphate salt.

HEPES contains N-donors and is not innocent with respect to silver(I) as shown by a crystal structure of a HEPES-silver(I) complex [21]. Two nitrogen atoms from the piperazine moieties of HEPES molecules as well as two oxygen atoms from the alcohol and sulfonate functions coordinate the silver ion in a distorted tetrahedral geometry. However, the binding affinity was not quantified.

Herein, we determined the affinity of HEPES for silver ions in order to quantize the buffer effect. In comparison, we also studied the effect of other buffers that were expected to possess the least interaction with silver ions (Scheme 1) to find out which one would be ideal for studies with silver(I) in biological media.

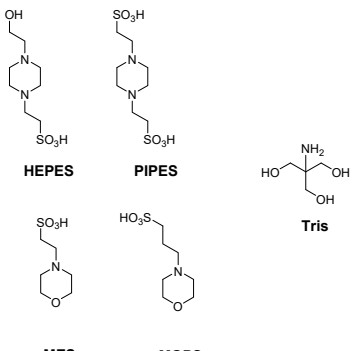

**Scheme 1.** Structures of buffers investigated for their affinities with silver ions.

## 2. Materials and Methods

Silver nitrate $AgNO_3$ was purchased from Carlo Erba reagents (RPE, Analytical 99+%). 4-(2-hydroxyethyl)-1-piperazineethane sulfonic acid (HEPES), 3-(*N*-morpholino)propanesulfonic acid (MOPS), tris(hydroxymethyl)aminomethane (Tris) (Roche), sodium nitrate $NaNO_3$ and potassium hydrogen phthalate (KHP) (Merck) were purchased from Sigma-Aldrich. Piperazine-1,4-bis(2-ethanesulfonic acid) (PIPES) and 2-(*N*-morpholino)ethanesulfonic acid (MES) were purchased from Roth. Nitric acid was purchased from Fluka and NaOH pellets from Acros. $HNO_3$ 0.1 M stock solution in 0.1 M $NaNO_3$ was standardized towards KHP (0.4 g) where the equivalence point is followed with the help of phenolphthalein indicator. NaOH 0.2 M stock solutions in 0.1 M $NaNO_3$ were standardized with stock solution of $HNO_3$ 0.1 M and used within two weeks to avoid carbonate formation. Buffers and silver nitrate were dissolved at a concentration of 0.05 M in 0.1 M $NaNO_3$. PIPES was insoluble in water, and NaOH had to be added up to a 0.069 M concentration (1.4 eq.).

Buffers were titrated manually in presence of 0.1 M $NaNO_3$ at 296 K over the pH range of 2–11 ($HNO_3$ was added to obtain the starting pH of 2) with NaOH 0.2 M as titrant. Changes in pH were monitored with a glass electrode (Primatrode with NTC Methrom, combined glass-Ag/AgCl electrode), calibrated daily with standard buffers at pH 4 and 7. Titrations were conducted in triplicates for each

buffer at three different concentrations between 2 and 12 mM with a sample volume of 50 mL. Silver nitrate was added at three different ratios from 0.2 to 1.0 equivalents compared to the buffer. The titration data were analyzed using the SUPERQUAD software according to equilibriums defined in Appendix A. Mass spectrometry was performed on an ESI-MS Bruker Esquire HCT in $H_2O$/MeOH solution (0.8:0.2) on the positive and negative mode with each buffer adjusted at pH 7 and 0.5 equivalent of silver nitrate.

## 3. Results

Acid dissociation constants were first determined without silver (Table 1, Figures 1 and S1) [22–25]. In the presence of silver nitrate, titrations were stopped at pH 8.0 because silver hydroxide and silver oxide are known to precipitate above this pH. The titration curve for HEPES with silver was found to have a lower plateau compared to HEPES alone, likely due to the coordination of HEPES to silver ions (Figure 1).

**Table 1.** Acid dissociation constants $pK_{an}$ ($n$= number of protons dissociated, see Figure S6 to visualize equilibrium considered) and complexation constants $\beta_{1,m}^{Ag,B}$ ($m$ = number of buffer molecules bound by one silver ion for the formation of the complex $[Ag(B)_m]$, see Figure S12 for proposed structures) obtained for the different buffers with potentiometric titrations and comparison with literature (L = HEPES, PIPES, MOPS, MES, Tris).

| Buffer | $pK_{an}$ (23 °C) [a] | $pK_{an}$ Literature (25 °C) | $\log(\beta_{1,m}^{Ag,B})$ (23 °C) [b] |
|---|---|---|---|
| HEPES | 7.46(1), 3.07(2) | 7.45(1) [23], 3.0(1) [24] | 2.36(2) |
| PIPES | 6.65(1), 2.54(2), 1.3(4) | 6.71(1) [23] | 1.95(3) |
| MOPS | 7.03(1) | 7.09(1) [23] | 1.1(1) |
| MES | 6.00(2) | 6.07(1) [23] | 1.69(8) |
| Tris | 8.24(1) | 8.08(1) [25] | 3.1(2), 6.5 (1) |

[a] Acid dissociation constants fitted on titration points between pH 2.0 and 11.5. [b] Stability constants fitted on titration points between pH 2.0 and 8.0.

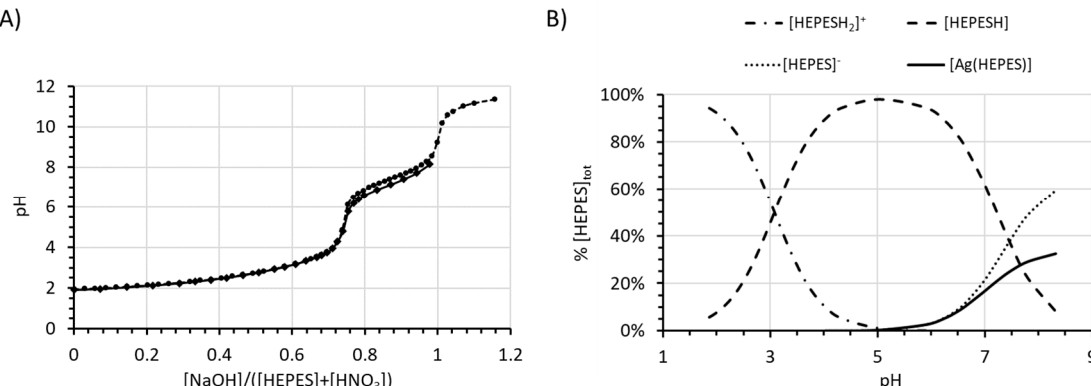

**Figure 1.** (**A**) Titration curves obtained for HEPES (7 mM) without (circles, dashed line) and in presence of silver nitrate (diamonds, plain line, 0.7 eq., 5 mM). (**B**) Speciation diagram according to pH for the species involving HEPES buffer.

We supposed the formation of a complex with one silver ion per HEPES ligand, based on the crystal structure obtained by Bilinovich et al. in 2011 resolved as a 1D coordination polymer with alternating HEPES and silver ions (Scheme 2) [21]. As solid-state structures do not always reflect the speciation in solution, three different ratios of silver to HEPES were tested. The titration curve fitted well (a 1) to the formation of a 1:1 complex and gave a stability constant of $\log(K_{1,1}^{Ag,HEPES})$ = 2.36(2) (Table 1). A stability constant for the formation of a hypothetical complex $[Ag(HEPESH)]^+$ with protonated HEPES in acid medium could be excluded as the fitting immediately results in negative

values when considering this equilibrium. Thus, the protonated complex [Ag(HEPESH)]$^+$ is unlikely to form in solution.

**Scheme 2.** Structure of HEPES in its neutral form and schematic representation of the crystal structure obtained in presence of silver(I).

Due to the non-negligible amount of silver bound to HEPES buffer, we investigated other buffers in the same way as well: another piperazine type buffer PIPES, and two morpholine type buffers MES and MOPS as well as Tris buffer (Figure S2–S5). Indeed, morpholinic and piperazinic families were selected to be the most innocent buffers because they contain bulky tertiary amines and a low number of other weakly coordinating groups (alcohols, sulfonates). Indeed, at physiological pH, these two families were considered to be suitable buffers due their weak complexation ability with other metals [12]. Tris buffer, which is widely used in biology, was expected to yield higher binding constants with silver ions due to the weak steric hindrance of the amine.

Stoichiometry of the complexes was proposed according to mass spectra and by testing various models for the determination of binding constants. Nevertheless, m/z signals in the positive and the negative modes for silver complexes were not observed for HEPES, PIPES, MES and MOPS buffers either because these are polymeric species or because the major species is neutral (Figures S7–S10). For the Tris complex, a 2:1 species was observed with two ligands around one metallic center [Ag(Tris)$_2$]$^+$ (Figure S11). For the determination of silver binding constants, larger errors were obtained when considering [Ag(MES)$_2$]$^-$ or [Ag$_2$(PIPES)] (Table S1) and negative values were found for [Ag(MOPS)$_2$]$^-$, so we decided to give only one stability constant for the formation of the [Ag(L)] complexes, with L = PIPES, MOPS, MES (Table 1 and Figure S12).

## 4. Discussion

Acid dissociation constants for the buffers alone were in good agreement with data from the literature (Table 1), confirming the validity of our measurements [22–25]. For the titration experiments with silver ions, the stability constant obtained for HEPES log($K_{1,1}^{\text{Ag,HEPES}}$) = 2.36(2) was lower than the value obtained for the 1:1 complex of HEPES with copper(II) log($K_{1,1}^{\text{Cu,HEPES}}$) = 3.22(2) [15]. This trend is expected as copper(II) is usually presenting greater affinities with nitrogen ligands due to its higher charge density [26,27].

Given the relatively high value for a buffer considered to be innocent of log($K_{1,1}^{\text{Ag,HEPES}}$) = 2.36(2) for the silver-HEPES complex, we simulated how binding constants of an analyte binding silver would be affected by the presence of HEPES buffer (Table 2, Appendix B). The decrease on stability constants that would be measured without taking the silver-HEPES complex into account depend on the concentration of the analyte and the relative stoichiometry with the buffer. However, these effects are still quite weak on the logarithmic scale of the stability constants, except when working at high concentrations, i.e., using Nuclear Magnetic Resonance (NMR) spectroscopy to obtain the stability constants.

**Table 2.** Apparent binding constants $\log(K_{app,1,1}^{Ag,L})$ corrected for the effect of buffer for various real values of binding constants $\log(K_{1,1}^{Ag,L})$ (L = peptide or analyte investigated for its complexation to silver, B = HEPES buffer at pH 7.4, 40 equivalents) and at different concentrations. Percentage of decrease is indicated in parenthesis.

| $\log(K_{1,1}^{Ag,L})$ | $\log(K_{app,1,1}^{Ag,L})$ | | | |
|---|---|---|---|---|
| | 6.4 | 4.0 | 3.0 | 2.0 |
| [L] = 10 μM, [B] = 0.4 mM | 6.38 (−0.3%) | 3.98 (−0.4%) | 2.98 (−0.6%) | 1.98 (−0.9%) |
| [L] = 100 μM, [B] = 4 mM | 6.25 (−2.3%) | 3.85 (−3.7%) | 2.85 (−4.9%) | 1.85 (−7.3%) |
| [L] = 500 μM, [B] = 20 mM | 5.92 (−7.5%) | 3.52 (−11.9%) | 2.52 (−15.9%) | 1.52 (−23.8%) |

In our previous study of SilE in presence of HEPES, the fact that HEPES binds to silver ions can be neglected. Binding constants of SilE model peptides with silver ions were indeed determined in presence of HEPES, but using a competitor with known binding affinity for silver ions. The competitor was then similarly affected by the buffer as the peptide ligand. The stability constant of the competitor was itself calculated in competition with imidazole (whose contribution to the thermodynamic equilibrium was taken into consideration).

Looking for evidence for the stoichiometry of the complexes formed with silver, published crystal structures were examined (Scheme 3) [28–30] but no structures were found for MOPS or Tris [31]. Triethanolamine buffer (TEOA) yields a $[Ag(TEOA)_2]^+$ complex, and this, together with the linear $[Ag(NH_3)_2]^+$ complex, suggests the possible formation of a complex $[Ag(Tris)_2]^+$ [32]. Interestingly, for the crystal structures of the silver-PIPES and silver-MES complexes, the silver(I) ions always has at least a coordination number of four (Table S2). The silver ion is typically maintained by two quite strong coordination bonds, preferentially with nitrogen atoms, and by two weaker secondary bonding interactions with oxygen atoms of sulfonate and alcohol groups. The silver-MES complex includes a benzimidazole ligand (Bz) together with the complexation of MES buffer. According to these structures, one could expect a 2:1 silver to buffer ratio for PIPES and a 1:2 ratio for MOPS, MES and Tris. The stoichiometry was confirmed by mass spectrometry for Tris buffer where the complex $[Ag(Tris)_2]^+$ was clearly identified as the main species in solution (Figure S11). Indeed, only the model with a 1:1 silver/buffer complex was working while fitting potentiometric data. Possible second binding constants are likely too weak to be precisely determined (Figure S12).

$[Ag_2(PIPES)(\mu_2\text{-}OH_2)]_n$ $[Ag(MES)(Bz)]_n$

**Scheme 3.** Crystal structures obtained for PIPES [28] and MES [29] buffers in presence of silver(I).

Unsurprisingly, the primary amine Tris is the strongest silver binder in this study and the stability constant obtained is comparable to other amine ligands such as ethanolamine [33–35]. This value is in line with other studies at different ionic strengths and temperatures that have been quantifying the interaction between Tris buffer and silver(I) [36,37], validating our approach. Morpholine type buffers were less coordinating than piperazine type buffers, as expected by previous results on unsubstituted morpholine [33] and piperazine [38] molecules. MOPS turned out to be clearly the least coordinating

buffer of the buffer series studied here (Figure S13). Compared to other metal ions, a lower first stability constant was obtained for silver (I) compared to the ones for copper(II), and similar to nickel(II) or cobalt(II) as found for amines in the literature [26,27,37,39].

To fully benefit from these results and apply them to the standard conditions of a titration (i.e., at constant pH, maintained with a buffer), stability constants were corrected to take into account the partial protonation of the buffer ligand (Figure 2A, Appendix C). The apparent binding constants are slightly decreased compared to the original values, especially when working at high concentrations. Please note that accurate determination of stability constants lower than $\log(K_{app,1,1}^{Ag,L}) = 3$ will ultimately necessitate the use of higher concentrations for the analyte and so for the buffer in order to see the association process. At these concentrations, and according to the third line of Table 2, buffer complexation cannot be neglected. Only high stability constants ($\log(K_{app,1,1}^{Ag,L}) \geq 4$) can thus be determined when using buffers. Another way to see the effect of the buffer on metal ion interactions is to calculate the amount of silver(I) ions bound to the buffer (Figure 2B, Appendix C). According to this percentage, a high proportion of silver ions -more than 90% for Tris buffer- would be complexed by the buffer. Fortunately, when measuring high stability constants at low concentrations for an analyte, the fact that silver ions are not free but bound to the buffer does not affect much the formation of the silver-analyte complex.

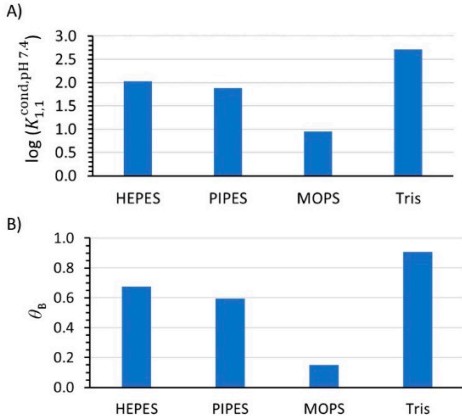

**Figure 2.** (**A**) Conditional (or apparent) stability constant for the complexation of silver(I) to the buffer at physiological pH 7.4 (for molecules comprised in their buffer range). (**B**) Fraction of silver bound to the buffer (total concentration of buffer 20 mM and silver 1 mM) at pH 7.4.

One could also decide to work at a lower pH (so MES could be considered, but not Tris, Figure S14 and S15B) or to work at different concentrations of buffers (Figure S15A). In the buffer range of the molecules studied here, whatever the conditions, MOPS was always the most suitable buffer. MES, HEPES and PIPES had similar coordination strength regarding silver(I) ions. They can reasonably be used if taking into consideration partial complexation to the buffer for accurate determination of stability constants of ligand/silver complexes.

In conclusion, between pH 6.5 and 7.9, MOPS would be recommended for the studies necessitating the use of silver(I) as it was the less coordinating buffer.

**Supplementary Materials:** The following are available online at http://www.mdpi.com/2624-8549/2/1/12/s1. Figure S1: Titration curves obtained for HEPES without and in presence of silver nitrate in solution, Figure S2: Titration curves obtained for PIPES without and in presence of silver nitrate in solution, Figure S3: Titration curves obtained for MOPS without and in presence of silver nitrate in solution, Figure S4: Titration curves obtained for MES without and in presence of silver nitrate in solution, Figure S5: Titration curves obtained for Tris without and in presence of silver nitrate in solution, Figure S6: Acid dissociation equilibriums considered in the present study for the different buffers, Figure S7: Mass spectra in positive and negative mode for HEPES buffer with silver nitrate, Figure S8: Mass spectra in positive and negative mode for PIPES buffer with silver nitrate, Figure S9: Mass spectra in positive and negative mode for MOPS buffer with silver nitrate, Figure S10: Mass spectra in positive and negative mode for MES buffer with silver nitrate, Figure S11: Mass spectra in positive mode for Tris buffer with silver nitrate, Figure S12: Proposed structures of complexes formed with silver. This stoichiometry was retained for determination of stability constants, Figure S13: Logarithm of stability constants for the first

complexation of silver(I) on ligands B (B= buffer studied in this paper), Figure S14: Logarithm of conditional (or apparent) stability constants for the first complexation of silver(I) on buffers at a fixed pH value pH = 6.7, Figure S15: Fraction of silver bound to the buffer, Table S1: Stability constants obtained when considering other equilibrium than the one for the formation of [Ag(L)] (complex [Ag$_2$(PIPES)] or [Ag(MES)$_2$]$^-$, Table S2: Bond distances (Ag-donor atom), average bond valences ($\nu_{Ag,N1X2}$ and $\nu_{Ag,O3-5}$) and total atom valence ($V_{Ag}$) in the molecular structures of [Ag$_x$(Buffer)$_m$].

**Author Contributions:** Conceptualization, L.B.; methodology, L.B.; validation, L.B.; and K.M.F.; formal analysis, L.B.; investigation, L.B. and S.B.-G.; resources, L.B.; data curation, L.B.; writing—original draft preparation, L.B.; writing—review and editing, L.B. and K.M.F.; visualization, L.B.; supervision, K.M.F.; funding acquisition, K.M.F. All authors have read and agreed to the published version of the manuscript.

**Funding:** This research was funded by Swiss National Science Foundation and BNF Universität Bern program.

**Acknowledgments:** I would like to acknowledge Jihane Hankache for mass spectra and Aurélien Crochet for search of crystallographic structures.

**Conflicts of Interest:** The authors declare no conflict of interest. The funders had no role in the design of the study; in the collection, analyses, or interpretation of data; in the writing of the manuscript, or in the decision to publish the results.

## Appendix A. Thermodynamic Equilibrium Used to Fit Potentiometric Titrations

$$\text{BufferH}_n \rightleftarrows \text{BufferH}_{n-1} + \text{H}^+ \quad K_{an} = \frac{[\text{BH}_{n-1}][\text{H}^+]}{[\text{BH}_n]} \tag{A1}$$

For acid dissociation constant $K_{an}$, there are one to three constants depending on the sum of amine group (one) and the number of sulfonates groups present in the buffer molecule.

A stability constant was then fitted with the fully deprotonated buffer according to Equation (A2).

$$\text{Buffer} \rightleftarrows [\text{Ag}(\text{Buffer})_m] + \text{H}^+ \quad \beta_{1,m}^{Ag,B} = \frac{[\text{AgB}_m]}{[\text{B}]^m[\text{Ag}]} \tag{A2}$$

For all buffers, $m = 1$ except in the case of Tris buffer where there are two constants for $m = 1$ and $m = 2$. For conversion between cumulative constants and stepwise constants (as usually found in literature):

$$\beta_{1,1}^{Ag,B} = K_{1,1}^{Ag,B} \text{ and } K_{1,2}^{Ag,B} = \beta_{1,2}^{Ag,B} / \beta_{1,1}^{Ag,B} \tag{A3}$$

The presence of a complex [Ag(BufferH)] was tested for the fitting of titration curves for all buffers but could not lead to any reliable results (constants were systematically negative). Thus, we consider that this complex was unlikely to be formed in solution.

## Appendix B. Calculation of Apparent Binding Constants of a Ligand Binding Silver

$$\text{Ag}^+ + \text{L} \rightleftarrows [\text{AgL}] \quad K_{1,1}^{Ag,L} = \frac{[\text{AgL}]}{[\text{L}][\text{Ag}]} \tag{A4}$$

We define an apparent binding constant which will be the one obtained if not considering the buffer-silver complexation:

$$K_{app,1,1}^{Ag,L} = \frac{[\text{AgL}]}{[\text{L}]\big([\text{Ag}]_{tot} - [\text{AgL}]\big)} = \frac{[\text{AgL}]}{\big([\text{L}]_{tot} - [\text{AgL}]\big)\big([\text{Ag}]_{tot} - [\text{AgL}]\big)} \tag{A5}$$

The mass balance equation for the total concentration of silver is expressed in Equation (A6):

$$[\text{Ag}]_{tot} = [\text{Ag}] + [\text{AgL}] + [\text{Ag}(\text{HEPES})] \tag{A6}$$

Concentration of silver complexes can be expressed according to the binding constants:

$$[Ag(HEPES)] = \frac{K_{1,1}^{Ag,B}}{1 + \sum_{i=1}^{n} \beta_{an}(10^{-pH})^n} \cdot \frac{[Ag][HEPES]_{tot}}{1 + \frac{K_{1,1}^{Ag,B}}{1 + \sum_{i=1}^{n} \beta_{an}(10^{-pH})^n} \cdot [Ag]} \tag{A7}$$

$$[AgL] = K_{1,1}^{Ag,L} \cdot \frac{[Ag][L]_{tot}}{1 + K_{1,1}^{Ag,L} \cdot [Ag]} \tag{A8}$$

Introducing Equations (A7) and (A8) in Equation (A6), we obtain an expression of total silver concentration as a function of silver free concentration [Ag].

$$[Ag]_{tot} = [Ag] + \frac{K_{1,1}^{Ag,B}}{1 + \sum_{i=1}^{n} \beta_{an}(10^{-pH})^n} \cdot \frac{[Ag][HEPES]_{tot}}{1 + \frac{K_{1,1}^{Ag,B}}{1 + \sum_{i=1}^{n} \beta_{an}(10^{-pH})^n} \cdot [Ag]} + K_{1,1}^{Ag,L} \cdot \frac{[Ag][L]_{tot}}{1 + K_{1,1}^{Ag,L} \cdot [Ag]} \tag{A9}$$

This concentration is optimized to minimize the difference between the actual concentration $[Ag]_{tot}$ and the one calculated by Equation (A9). Once the concentration of free silver [Ag] at hand, the apparent binding constant can be calculated from Equation (A8) and reintroducing in Equation (A5).

**Appendix C. Calculation of Conditional Stability Constants at a Certain pH and Calculation of Percentage of Metal Bound to the Buffer $\theta_B$**

Conditional stability constants are defined as the apparent binding constants of the complex between silver(I) and the buffer B at a certain pH value. Thus, we considered that a certain part of the buffer is not coordinating silver(I) as it is protonated but it is still considered in the equilibrium as shown in Equation (A10):

$$K_{1,1}^{cond,pH\ cst} = \frac{[AgB]}{[Ag]\left([B] + \sum_{i=1}^{n}[BH_n]\right)} = \frac{K_{1,1}^{Ag,B}}{1 + \sum_{i=1}^{n} \beta_{an}(10^{-pH})^n} \tag{A10}$$

For the calculation of the concentration of species and the percentage of metal bound to the buffer, we first established the mass balance equations:

$$[B]_{tot} = [B] + \sum_{i=1}^{m} m[AgB_m] + \sum_{i=1}^{n}[H_nB] \tag{A11}$$

$$[B]_{tot} = [B] + \sum_{i=1}^{m} m[AgB_m] + \sum_{i=1}^{n}[H_nB] \tag{A12}$$

Then we rearrange Equation (A6) and silver total concentration to express the concentration of free silver:

$$[Ag] = \frac{[Ag]_{tot}}{1 + \sum_{i=1}^{m} \beta_{1,m}^{Ag,B}[B]^m} \tag{A13}$$

Free concentration of silver was then reintroduced in Equation (A11):

$$[B]_{tot} = [B]\left(1 + \sum_{i=1}^{m} \frac{m \cdot \beta_{1,m}^{Ag,B}[Ag]_{tot}[B]^{m-1}}{1 + \sum_{i=1}^{m} \beta_{1,m}^{Ag,B}[B]^m} + \sum_{i=1}^{n} \beta_{an}[H]^n\right) \tag{A14}$$

We assumed a certain value for the concentration of the free ligand [B] to obtain a value of $[B]_{tot, calc.}$ with Equation (A14) at a certain pH value. Difference between the calculated total ligand concentration and the one set in the experiment was minimized by tuning the value of free ligand [B].

Once the parameter of free ligand/buffer [B] has been optimized, one could calculate the concentration of complexed species $[AgB_m]$ with concentration of free metal being determined with Equation (A12):

$$\left[AgB_m\right] = \beta_{1,m}^{Ag,B}[Ag][B]^m \tag{A15}$$

The percentage of metal bound to the buffer is then calculated according to Equation (A16):

$$\theta_B = \frac{\sum_{i=1}^m \left[AgB_m\right]}{[Ag]_{tot}} = \frac{\sum_{i=1}^m \beta_{1,m}^{Ag,B}[Ag][B]^m}{[Ag]_{tot}} \tag{A16}$$

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
