# Peer review of "Appropriate Buffers for Studying the Bioinorganic Chemistry of Silver(I)†"

_chemistry, doi:10.3390/chemistry2010012_

Round 1
Reviewer 1 Report
This is an interesting contribution studying selected buffers for in presence of silver nitrate. By means of potentiometric titrations, Fromm and coworkers have determined stability constants for the formation of [Ag(Buffer)] complexes and presenet a sound piece of work.
Accept as the paper stands.
Author Response
Response to Reviewer 1 Comments
Thank you for reviewing our article.
Reviewer 2 Report
The paper by Babel et al. entitled: “Appropriate buffers for studying the bioinorganic chemistry of silver(I)” is well designed and conducted analytical work. As Authors claimed, these studies are necessary to work in buffer systems that are best suitable, i.e. that are the least interacting with silver cations. Selected buffers such as HEPES, PIPES, MOPS, MES and TRIS were therefore investigated for their use in presence of silver nitrate. These results can be very helpful for researchers who investigate silver(I) antimicrobial properties in biological systems.
I have two question/suggestion to resolve:
- The obtained acid dissociation constants pK, included in the Table 1, should be assigned to the appropriated groups in buffers, capable of deprotonation.
- Is it possible to show proposed silver coordination modes to all investigated buffers? The Authors have shown only crystal structures presented in the literature, but what about structures in the solution system?
Author Response
Response to Reviewer 2 Comments
Point 1: The obtained acid dissociation constants pK, included in the Table 1, should be assigned to the appropriated groups in buffers, capable of deprotonation.
We have added a Figure in the Supplementary Material (S6) to describe the acid dissociation equilibrium considered in this study.
Point 2: Is it possible to show proposed silver coordination modes to all investigated buffers? The Authors have shown only crystal structures presented in the literature, but what about structures in the solution system?
We have added a Figure in the supplementary material (S12) for the proposed silver coordination. To answer the second question, we have added mass spectra to try to have an information on the stoichiometry in solution. Other methods to get some knowledge about the stoichiometry in solution were not considered due to the uncertainty regarding Job plots (see Hibbert, D. B., Thordarson, P., Chem. Commun. 2016), especially if the second binding constant is lower than the first one.
Reviewer 3 Report
The manuscript reported by Babel and co-workers describes the complex formation processes between silver(I) and several common buffers. My opinion is that the topic is interesting and important with respect to practical applications. Therefore, the manuscript should be published after considering the following remarks.
- Since it seems, that the concentration of AgNO3 stock solution is directly calculated from the weight of the salt and the volume of the stock solution, without any further check. Thus, the purity and the brand of the salt should be reported in the Materials and Methods.
- The complex formation processes should be described in more detail. It is clear from Figure 1. that the complexation between Ag(I) and HEPES starts around pH 6, however, a calculated distribution of the complexes shows better the evolution of complex formation as a function of pH. In Figure 1, the metal to buffer ratio and the concentration of the buffer should be indicated.
- Discrepancies between the manuscript and Supplementary. In the manuscript, only the Ag(Tris) complex is reported, however, the Supplementary deals with the Ag(Tris)2 complex, too. This point should be clarified. Moreover, the fitted pH range should be reported in Table 1.
- The authors used only pH-potentiometric titration to determine the stability constants of the silver buffer complexes. To confirm the equilibrium model, further experiments should be carried out. Since silver has diamagnetic behavior, NMR experiment offer the possibility to justify the species formed in aqueous solution.
- Table 2 did not show correctly the influence of the buffer on the stability of the apparent binding constants. It is clear that the increased concentration of the buffer results in higher deviation between the calculated and corrected binding constants which is due to the competition for the metal binding. To better compare the binding constants and the influence of the buffer, authors should use different ligand concentrations but the same buffer concentration.
- It is difficult to follow the titration curves. The authors should report the titration curves by use of the base equivalent relative to the ligand. Such representations of the titration curves show better the stoichiometry and the additional base consumption processes.
Author Response
Response to Reviewer 3 Comments
Point 1: Since it seems, that the concentration of AgNO3 stock solution is directly calculated from the weight of the salt and the volume of the stock solution, without any further check. Thus, the purity and the brand of the salt should be reported in the Materials and Methods.
The brand was already specified, but we have added the type and the purity of the salt.
Point 2: The complex formation processes should be described in more detail. It is clear from Figure 1. that the complexation between Ag(I) and HEPES starts around pH 6, however, a calculated distribution of the complexes shows better the evolution of complex formation as a function of pH. In Figure 1, the metal to buffer ratio and the concentration of the buffer should be indicated.
We have added a speciation diagram and the metal to buffer ratio in the legend of Figure 1 according to this comment.
Point 3: Discrepancies between the manuscript and Supplementary. In the manuscript, only the Ag(Tris) complex is reported, however, the Supplementary deals with the Ag(Tris)2 complex, too. This point should be clarified. Moreover, the fitted pH range should be reported in Table 1.
We have added a short part to explain these discrepancies (see alo answer for the next point). Acid dissociation constants were recorded from pH 2.0-11.5. Silver binding constants between pH 2.0-8.0 as now stated in the Table.
Point 4: The authors used only pH-potentiometric titration to determine the stability constants of the silver buffer complexes. To confirm the equilibrium model, further experiments should be carried out. Since silver has diamagnetic behavior, NMR experiment offer the possibility to justify the species formed in aqueous solution.
I agree that NMR would be perfectly suited for the determination of binding constants regarding the use of diamagnetic metal and the magnitude of the binding constant. Nevertheless, NMR titrations in deuterated water requires the use of a buffer so that pH is not changing upon addition of silver. In our case, binding of the buffer and addition of Lewis acid silver nitrate will ultimately lead to changes in pH (ΔpH≈1.0 taking silver concentrations between 0.5 and 50 mM and a pKa for silver dissociation of 11.7). The change of pH together with the silver complexation will result in shifts of the proton signals. I do not know how the reviewer want us to address this issue. In fact, the competition with protonation equilibrium is a problem regarding any other kind of titration method which necessitates to record the pH as well as recording another physical data along the titration. Thus, a titration would require the use of a small apparatus for NMR (small pH electrode) as well as single titration point (no additive titration, pH adjusted for each point). Errors on concentration could not be avoided by dipping the NMR-type electrode each time we want to record the pH. For these reasons, I do not think this time-consuming experiment is justified in our case. Moreover, most examples in the literature reporting binding constants only report one type of titration method. To prove our rigoristic approach, we have added mass spectra to get some knowledge about the stoichiometry in solution.
Point 5: Table 2 did not show correctly the influence of the buffer on the stability of the apparent binding constants. It is clear that the increased concentration of the buffer results in higher deviation between the calculated and corrected binding constants which is due to the competition for the metal binding. To better compare the binding constants and the influence of the buffer, authors should use different ligand concentrations but the same buffer concentration.
A single buffer concentration would require the use of the largest concentration of buffer. The same deviation would be observed whatever is the concentration of the ligand because the amount of silver bound to the buffer is depending only on the buffer concentration. The table was considering typical concentrations used for fluorescence or NMR titrations. According to your comment, we decided to change one line of our table to consider medium concentrations 100 uM and keep the buffer concentration to 40 equivalents.
Point 6: It is difficult to follow the titration curves. The authors should report the titration curves by use of the base equivalent relative to the ligand. Such representations of the titration curves show better the stoichiometry and the additional base consumption processes.
We have changed spectra according to this comment and titration curves are represented according to the base equivalent relative to the amount of acid present in the titration (which is usually the ligand and the nitric acid concentration).
Round 2
Reviewer 3 Report
The revised manuscript adequately reflects most of the comments and I recommend the paper for publication. I have only one minor comment.
The NMR titrations in the presence and absence of silver can be performed using water suppression sequences. In this case, the solution can be prepared in light water and the D2O can be added to the sample in capillary. Consequently, the titrations must not be performed in the NMR tube, thus the errors on pH reading and concentrations could be avoided.